# Acknowledging the role of patient heterogeneity in hospital outcome reporting: Mortality after acute myocardial infarction in five European countries

Micaela Comendeiro-Maaløe[1,2], Francisco Estupiñán-Romero[1,2], Lau Caspar Thygesen[3], Céu Mateus[4], Juan Merlo[5], Enrique Bernal-Delgado[1,2]*, on behalf of the ECHO consortium¶

1 Health Services and Policy Research Group, Institute for Health Sciences in Aragon (IACS), Zaragoza, Spain, 2 Network for Health Services Research in Chronic Patients (REDISSEC), Madrid, Spain, 3 National Institute of Public Health, University of Southern Denmark, Copenhagen, Denmark, 4 Division of Health Research, Lancaster University, Lancaster, England, United Kingdom, 5 Unit for Social Epidemiology, Sweden & Centre for Primary Health Care Research, Region Skåne, Faculty of Medicine, Lund University Malmö, Malmö, Sweden

¶ Membership of the ECHO Consortium is listed in the Acknowledgments.
* ebernal.iacs@aragon.es

## Abstract

### Background

Hospital performance, presented as the comparison of average measurements, dismisses that hospital outcomes may vary across types of patients. We aim at drawing out the relevance of accounting for patient heterogeneity when reporting on hospital performance.

### Methods

An observational study on administrative data from virtually all 2009 hospital admissions for Acute Myocardial Infarction (AMI) discharged in Denmark, Portugal, Slovenia, Spain, and Sweden. Hospital performance was proxied using in-hospital risk-adjusted mortality. Multi-level Regression Modelling (MLRM) was used to assess differences in hospital performance, comparing the estimates of random intercept modelling (capturing hospital general contextual effects (GCE)), and random slope modelling (capturing hospital contextual effects for patients with and without congestive heart failure -CHF). The weighted Kappa Index (KI) was used to assess the agreement between performance estimates.

### Results

We analysed 46,875 admissions of AMI, 6,314 with coexistent CHF, discharged from 107 hospitals. The overall in-hospital mortality rate was 5.2%, ranging from 4% in Sweden to 6.9% in Portugal. The MLRM with random slope outperformed the model with only random intercept, highlighting a much higher GCE in CHF patients [VPC = 8.34 (CI95% 4.94 to 13.03) and MOR = 1.69 (CI95% 1.62 to 2.21) vs. VPC = 3.9 (CI95% 2.4 to 5.9), MOR of

**Data Availability Statement:** The access to pseudoanonimised patient-level administrative data (patient episodes: hospital admissions and discharges) supporting the manuscript findings is restricted to avoid data misuse in accordance with the System Level Security Policy of the Unit for Health Services and Policy Research at the Institute for Health Sciences in Aragon (IACS) (Available at: www.atlasvpm.org/grupo-coordinador/ARiHSPinformationsecuritypolicy) within the framework of the Spanish legal system, in particular, Regulation (EU) 2016/679, the Law 3/2018 on Personal Data Protection, the Law 14/2007 on Biomedical Research and the Law 37/2007 and Law 18/2015 on the Public Sector Information Reuse. Consequently, we have specified that only aggregated data may be accessed, upon request, throughout our data sharing agreement, by contacting the senior author of the manuscript Enrique Bernal-Delgado (ebernal.iacs@aragon.es) and/or the legal officer Ramón Launa-Garcés (rlaunag.iacs@aragon.es).

**Funding:** This study has been funded by the Spanish Institute for Health Carlos III through the project "The Research Network on Health Services Research on Chronic Patients (REDISSEC)", registered grant RD16/0001/0007, and co-funded by European Regional Development Fund "Investing in your future". On the other hand, the ECHO Project received funding from the 7th Framework Programme of the European Union (2010-2014) with Grant HEALTH-F3-2010-242189 and from the European Union's Health Programme (2014-2020) with Grant 664691/BRIDGE Health. Additionally, the authors Enrique Bernal-Delgado, Micaela Comendeiro-Maaløe and Francisco Estupiñán-Romero have been partially funded by the Spanish Institute for Health Carlos III through a public competitive grant (RD16/0001/0007) as research members of "The Research Network for Health Services Research on Chronic Patients (REDISSEC)". All these funding sources played no role in the study design, data collection or the analysis, decision to publish, or preparation of the manuscript. There was no additional external funding received for this study.

**Competing interests:** The authors have declared that no competing interests exist.

1.42 (CI95% 1.31 to 1.54) without CHF]. No agreement was observed between estimates [KI = -0,02 (CI95% -0,08 to 0.04].

## Conclusions

The different GCE in AMI patients with and without CHF, along with the lack of agreement in estimates, suggests that accounting for patient heterogeneity is required to adequately characterize and report on hospital performance.

## 1. Introduction

The growing availability and use of administrative data are resulting in a profusion of health-care performance assessment initiatives worldwide. Either institutionally framed or developed under the umbrella of research projects, the wealth of administrative data offers the opportunity to access larger samples of patients, covering virtually all providers in a health plan, allowing cross-country comparisons and most importantly, enabling the systematic and continuous monitoring of providers' performance. Many institutional-based [1–7] and research-oriented examples [8–15] illustrate this enormous potential. On the other hand, as performance assessment is increasingly deemed to be the basis for different value-based initiatives (e.g. benchmarking strategies, pay for performance schemes, patient choice programs, etc), decision makers are increasingly calling for trustworthy measurements and reliable reporting [16].

In this respect, analytical methods play a critical role. Once the use of ordinary (single level) regression models were shown to be inappropriate, as they circumvent the interdependence of patient outcomes within a hospital (i.e. patient risk within a hospital is more alike than patient risk from a different hospital) [9, 12, 15–19], and are at risk of the Yule-Simpson paradox [20], marginal models (Generalized Estimating Equations, GEE) or multilevel modelling (MLRM) have become increasingly popular, although their approach and interpretation are clearly different; while the use of GEE focus on the estimation of the population-averaged risk of death adjusting hospitals' heterogeneity, MLRM assumes that each hospital has their own underlying risk of an event, and this risk varies across hospitals (i.e. the probability of an event is conditional to the place where the patient is treated). Accordingly, MLRM has been suggested as a more appropriate approach when hospital-specific interpretations are needed [21].

But most importantly, variations in hospital performance are usually presented as the comparison of adjusted average measures, excluding the possibility that hospital performance may also be conditioned by patient heterogeneity, for example, determining the care responses to specific subgroups of patients [22]. One fundamental feature of MLRM in hospital performance assessment is that MLRM can drop the assumption that the underlying risk for an individual is the same for all hospitals, allowing this risk to vary at hospital level; therefore, the hospital effect also becomes a function of patient heterogeneity [23]. In practical terms, this property, which implies the inclusion of random slopes, allows the development of specific performance measurements for subgroups of patients. Therefore, the observation of better or worse performance will refer not just to the hospital outcome obtained for the regular patient but also to the hospital achievement for specific subgroups of individuals. This well-known property of MLRM has scarcely been exploited in the assessment and reporting of hospital performance.

In this paper, we use MLRM to draw out the relevance of accounting for patient heterogeneity when reporting on hospital performance, using in-hospital mortality in acute myocardial

infarction as a case study. Including a random slope for AMI patients with coexistent CHF will show the relevance of accounting for patient heterogeneity in hospital performance reporting.

## 2. Methods

### 2.1. Design, population and setting

An observational cross-sectional study utilising administrative data representing virtually all hospital admissions for AMI in patients aged from 40 to 80, treated in 434 hospitals from 5 European countries (Denmark, Portugal, Slovenia, Spain and Sweden) in 2009, totalling 73,812 potential discharged episodes. Hospitals accounting for fewer than 250 AMI episodes in 2009 (discretionary threshold) were excluded from the sample in order to reduce structural heterogeneity across hospitals and gain robustness in the estimations (Fig 1). The final sample accounted for 107 hospitals, accounting for 46,875 AMI episodes (63.5% of all assisted episodes), from which 5.2% deceased (2,451 case-fatalities). CHF coexisted in 13,5% of the sample (6,314 AMI episodes).

### 2.2. Endpoints

Our work comprised two consecutive endpoints; firstly, the variation in the hospital effect (i.e. GCE) when including a random slope for CHF patients in the MLRM; and, additionally, the level of agreement in hospital outcomes, contrasting both types of hospital GCE (i.e. under the assumption that the underlying risk for an individual level association is the same for all the hospitals or under the assumption that the underlying risk for CHF patients varies across hospitals).

### 2.3. Variables in the models

As aforementioned, the hospital outcome in this study (i.e. proxy of performance measure) was the adjusted in-hospital mortality risk in AMI patients who stayed for up to 30 days after admission; thus, inpatients with admission diagnosis code 410* in those countries using ICD-MC 9th (Spain and Portugal) and I21* and I22* in those countries using ICD 10th (Denmark, Slovenia and Sweden). Those admissions due to pregnancy, puerperium or childbirth were excluded (codes ICD-MC 9th O00*-O99* or ICD10th 630–677). [detailed in S1 Appendix]

The patient-level independent variables were: a) *age*, categorized as 40–49, 50–59, 60–69 and 70–80, using the youngest group as the reference group; b) *sex*, using male as the reference

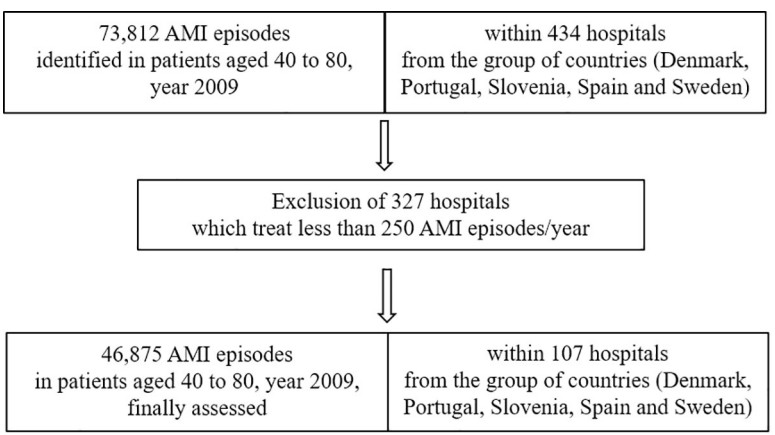

**Fig 1. Study population.** Flow diagram showing the episodes with an Acute myocardial infarction diagnosis according to hospital selection.

category; c) patient comorbidities, computed as an Elixhauser risk score [24, 25], obtained from the predicted probability of death for each of the episodes modelled with a single level logistic regression; and d) the coexistence of congestive heart failure (CHF) in the episode of AMI. Patients with CHF constituted the subgroup of patients of interest, being more fragile than the regular patient and supposedly with the requirement for a higher intensity of care. A CHF was flagged when ICD9th codes 398.91 and 428*, and ICD10th code I50*, were found in any secondary diagnosis recorded within the same episode. The definitions and corresponding codes were developed and validated in the context of the ECHO project [26]. When it comes to the hospital-level, no specific variables where included except the GCE captured as a random-effect. Finally, a dummy variable identifying the country of admission was included using Sweden as a reference in the comparisons.

## 2.4. Analyses

Upon the estimation of the basal risk of death associated to patient features and country of residence (basal model) throughout a conventional single level logistic regression model including age, sex, the Elixhauser score of risk, the coexistence of CHF, and the country of "treatment" (see variable definitions above), two MLRM models were built to estimate the hospital-specific risk of death for patients with AMI. For that purpose, we followed the methodology described elsewhere in a two-stage process [15].

The first MLRM specification included a random intercept for the hospital level in a two-level multilevel logistic regression model, so that each hospital got its own intercept (i.e. basal risk of death) (Eq 1).

$$ln\left(\frac{\pi_{ij}}{1 - \pi_{ij}}\right) = \gamma_{00} + \sum_{n=1}^{N}\gamma_{nj}X_{nj} + \sum_{k=1}^{K}\gamma_{kj}D_j + \gamma_{1j}Z_{ij} + \gamma_{2j}CHF_{ij} + u_{oj} + \varepsilon_{ij} \quad (1)$$

Where $u_{oj} + \varepsilon_{ij}$ is the random effect part of the model

$$u_{oj} \sim iid\ N(0, \sigma_u^2)$$

$$\varepsilon_{ij} \sim iid\ N(0, \sigma_\varepsilon^2)$$

The second MLRM specification, as an extension of the previous one, included a random slope, allowing each hospital to vary their risk slope according to a specific group of patients (in our case, patients with CHF). In practice, we obtain a hospital variance for patients without CHF and a different hospital variance for patients with CHF (see Eq 2).

$$ln\left(\frac{\pi_{ij}}{1 - \pi_{ij}}\right) = \gamma_{00} + \sum_{n=1}^{N}\gamma_{nj}X_{nj} + \sum_{k=1}^{K}\gamma_{kj}D_j + \gamma_{1j}Z_{ij} + \gamma_{2j}CHF_{ij} + u_{oj} + u_{2j}CHF_{ij} + \varepsilon_{ij} \quad (2)$$

Where
$X_{nj}$ are the N variables characterising the gender and age of patients
$Z_{ij}$ is the probability of death for a patient according to the concurrence of Elixhauser comorbidities, except CHF

$$Z_{ij} = ln\left(\frac{p_i}{1 - p_i}\right) = \gamma_{00} + \sum_{k=1}^{K}\gamma_{k0}x_{ki} + \varepsilon_i$$

$D_j$ are dichotomous variables which identify the countries where hospitals belong

$u_{oj} + u_{2j} CHF_{ij} + \varepsilon_{ij}$ is the random effect part of the model

$$\begin{pmatrix} u_{oj} \\ u_{2j} \end{pmatrix} \sim iid\, N\left(0, \Omega_u^2\right), \; \Omega_u = \begin{pmatrix} \sigma_{u0}^2 \\ \sigma_{u0}^2 \sigma_{u2}^2 \end{pmatrix}$$

$$\varepsilon_{ij} \sim iid\, N(0, \sigma_\varepsilon^2)$$

**2.4.1. Estimation of the general contextual effect.** The GCE was estimated for both models, the random intercept model and the extended model which adds a random slope. For both models, the hospital variance derivatives, the Variance Partition Coefficient (VPC) and the Median Odds Ratio (MOR), were also calculated.

(i) We calculated the VPC based on the latent response formulation of the model as [21, 22, 27]:

$$VPC = \frac{\sigma_u^2}{\sigma_u^2 + \frac{\pi^2}{3}}$$

Where $\sigma_u^2$ denotes the hospital variance, and $\frac{\pi^2}{3}$ the variance of a standard logistic distribution ($\pi = 3.1416$).

VPC is reported as a percentage that goes from 0% to 100%. If hospital differences (i.e. variance) were not relevant for understanding the individual differences in the latent propensity of death, the VPC would be 0%. That is, the hospitals would be similar to random samples taken from the whole patient population.

(ii) The median odds ratio (MOR) is an alternative interpretation of the magnitude of hospital variance [28–30]. The MOR is defined as the median value of the distribution of odds ratios (OR) obtained when randomly picking two patients with the same covariate values from two hospitals with a different underlying risk of an event of interest, and comparing the one from the hospital with the higher risk to the one from the hospital with the lower-risk. In simple terms, the MOR can be interpreted as the median increased odds of reporting the outcome if a patient is treated in another hospital with a higher risk. The MOR is calculated as:

$$\exp\left(\sqrt{2\sigma_u^2}\, \Phi^{-1}(0.75)\right)$$

where $\Phi^{-1}(\cdot)$ represents the inverse cumulative standard normal distribution function. In the absence of hospital variation (i.e. $\sigma_u^2 = 0$), the MOR is equal to 1. Theoretically, the MOR values may extend from 0 to $\infty$ and the higher the MOR value, the more relevant the hospital effect in terms of patient outcome. The MOR translates the hospital variance estimated on the log-odds scale to the widely used OR scale, making MOR values comparable to the individual OR covariates in the model.

For the estimation of the models, we used the Restricted Iterative Generalized Least Squares (RIGLS) method to obtain the values needed to finally run the Markov Chain Monte Carlo (MCMC) estimation method [31]. The goodness-of-fit of the models was assessed through the Bayesian Diagnostic Information Criterion (BDIC).

We performed the analyses using MLwiN run on Stata® statistical software: Release 13, College Station, TX: StataCorp LP and MlwiN version 2.35, The Centre for Multilevel Modelling, University of Bristol [32].

**2.4.2. Concordance in hospital performance.** Finally, for the assessment of concordance (i.e. agreement in hospital performance on patients with and without CHF), we compared the residuals from both random parts in the extended model, the intercept $[u_{oj}]$ and the slope

$[u_{2j}]$. The level of concordance between both residuals was studied using a measurement of agreement between observers for categorical variables. As the number of cases per country varied substantially, a weighted Kappa Index was estimated [33]. The choice of this estimator depends on how commonly performance measurements are reported through funnel plots, so units of analysis are categorized as: hospitals with residuals which are statistically above the average (i.e. exhibiting a higher risk of death than expected), hospitals with residuals which are statistically below the average (i.e. exhibiting a lower risk of death than expected), and hospitals that did not differ statistically from the expected risk of death, irrespective of their actual position above or below (i.e. hospitals within the funnel boundaries). According to this approach, hospitals in the sample were classified into three possible situations: better, neutral or worse than the expected, this categorization becoming the subject of the concordance measurement. As for interpretation purposes, the higher the Kappa Index, the higher the concordance between the two estimated hospital effects, which could suggest that hospitals perform equally in the patients without CHF as in the patients with CHF. Conversely, low concordance could suggest that hospitals perform differently.

## 2.5. Data sources

Hospital admissions from Denmark, Portugal, Slovenia and Spain were extracted from the database consolidated and validated during the ECHO project [30]. In turn, the Swedish Patient Register [34, 35] provided the hospital data for Sweden. Both pseudonymised datasets were linked into a single database, stored, validated and analysed in a secure server set up in the premises of the Faculty of Medicine at Lund University (Malmo, Sweden), as foreseen in the access policies of the Swedish Register.

## 2.6. Ethics statement

This study, observational in design, used retrospective anonymized, non-identifiable and non-traceable data, and was conducted in accordance with the amended Helsinki Declaration, the International Guidelines for Ethical Review of Epidemiological Studies, and Spanish laws on data protection and patients' rights. The study implies the use of pseudonymised data, using double dissociation (i.e. in the original data source and once the data are stored in the database for analysis) which actually impedes patients' re-identification. The information supplied for the European collaboration presented the same strong characteristics of confidentiality as the other collaborating countries.

## 3. Results

The final sample was composed of 46,875 episodes with a primary admission diagnosis of AMI, discharged from 107 hospitals. Overall, 6,314 patients underwent a concomitant CHF. By countries, Denmark treated 4,635 of those AMI episodes in 6 hospitals (9.9% of the episodes in the sample); Portugal accounted for 6,217 from 16 hospitals (13.3% of the episodes), while Slovenia yielded 1,898 episodes in 3 of the hospitals (4.1% of the admissions analysed). In turn, Spain treated 23,043 AMI episodes in 56 hospitals (49.2% of the episodes in the sample) while Sweden dealt with 11,082 of the AMI episodes in 26 hospitals (23.6% of the episodes).

The sample had 38.2% of patients aged 70 to 80, varying across countries, with 33.2% in Denmark and 42.5% in Sweden. Overall, 26.4% of the patients were female, ranging from 23.6% in Spain to 30.6% in Sweden. The average risk score (i.e. predicted probability of death according to the Elixhauser comorbidities) for the whole sample was 5.2, ranging from 4.5 in

**Table 1. Description of the study sample, per country (2009).**

|  | DNK | PRT | SVN | ESP | SWE | TOTAL |
|---|---|---|---|---|---|---|
| **AMI patients (n)** | 4,635 | 6,217 | 1,898 | 23,043 | 11,082 | 46,875 |
| **Age distribution** |  |  |  |  |  |  |
| % patients in age group 40–49 | 11.28% | 11.57% | 10.75% | 12.51% | 6.41% | 10.75% |
| % patients in age group 50–59 | 22.37% | 22.33% | 26.45% | 22.21% | 17.08% | 21.20% |
| % patients in age group 60–69 | 33.12% | 28.55% | 27.34% | 27.65% | 34.03% | 29.80% |
| % patients in age group 70–80 | 33.23% | 36.56% | 35.46% | 37.63% | 42.48% | 38.24% |
| **Sex**—Percentage of women | 26.11% | 28.57% | 29.72% | 23.65% | 30.64% | 26.44% |
| **Risk Score (mean)** | 4.51 | 5.77 | 5.55 | 5.47 | 4.67 | 5.23 |
| minimum | 4.16 | 4.16 | 4.16 | 4.16 | 4.16 | 4.16 |
| maximum | 68.09 | 73.04 | 57.75 | 76.85 | 68.34 | 76.85 |
| **CHF patients per hospital (mean)\*** | 78 | 56 | 79 | 58 | 57 | 59 |
| minimum | 38 | 19 | 35 | 13 | 28 | 13 |
| maximum | 114 | 141 | 117 | 197 | 139 | 197 |
| **Deceased (n)** | 206 | 432 | 84 | 1,286 | 443 | 2,451 |
| **In-hospital Crude Mortality Rate** (per 100 patients at risk in 2009) | 4.82 | 6.91 | 4.16 | 5.63 | 3.99 | 5.34 |
| minimum | 2.83 | 3.72 | 1.77 | 0.46 | 1.11 | 0.46 |
| maximum | 10.30 | 13.09 | 7.17 | 9.68 | 7.12 | 13.09 |
| **HOSPITALS** | 6 | 16 | 3 | 56 | 26 | 107 |
| Episodes per hospital (mean) | 772 | 389 | 633 | 412 | 426 | 438 |
| minimum | 505 | 254 | 283 | 258 | 250 | 250 |
| maximum | 951 | 604 | 1015 | 753 | 1054 | 1054 |

\* CHF stands for Congestive Heart Failure

Denmark to 5.8 in Portugal. Finally, the overall proportion of AMI patients with congestive heart failure was 59%, ranging from 56% in Portugal to 79% in Slovenia (Table 1).

Overall, 5.3 per 100 AMI patients died in hospital (2,451 cases out of 46,875) in the period of study; the crude mortality rate ranged from 0.5 to 13.1 per 100 patients at risk, for an interquartile interval of 1.63. By countries, Sweden, Slovenia and Denmark showed the lowest in-hospital mortality rates, 4, 4.2 and 4.8 per 100 patients at risk respectively, while Portugal showed the highest with 6.91 per 100 patients at risk, followed by Spain with an in-hospital mortality rate of 5.6 per 100 patients at risk (Table 1).

Table 2 shows the estimated adjusted-risks of death in the basal model, the basic GCE and the extended RS model. As observed in the basal model, the AMI risk of death increased with age (as compared to patients younger than 50), with the highest risk amongst the oldest (4.8 times more likely to die), the presence of comorbidities (2.1 times more likely), and the coexistence of CHF (2.8 times more likely). As compared to Sweden, patients living in Portugal were at 78% more risk of death, Denmark and Spain showing a 36% increased risk, while Slovenia barely registered a 6% increase. Being female did not increase the risk of death. Patient-level and country-level estimates were similar in both MLRM (second and third column in Table 2).

Both MLRM models confirmed the existence of a GCE; thus, beyond individuals' features, we observed an increase in the risk of death associated to the hospital of treatment. Moreover, in the specific case of the extended model with a random slope (the best model according to BDIC), the GCE was much higher in CHF patients, [VPC of 8.34 (CI95% 4.94 to 13.03) and a MOR value of 1.69 (CI95% 1.62 to 2.21)] than in those without CHF [VPC = 3.9 (CI95% 2.4 to 5.9), MOR of 1.42 (CI95% 1.31 to 1.54)].

**Table 2. Factors associated to in-hospital mortality in AMI patients (2009).**

|  | Basal model (single level) | MLRM (random intercept) | MLMR (random slope) |
|---|---|---|---|
| **Gender** | | | |
| Male | Ref | Ref | Ref |
| Female | 1.09 | 1.08 | 1.08 |
|  | (0.99 1.19) | (0.98 1.18) | (0.98 1.18) |
| **Age** | | | |
| 40–49 | Ref | Ref | Ref |
| 50–59 | 1.68 | 1.71 | 1.70 |
|  | (1.29 2.19) | (1.33 2.25) | (1.34 2.12) |
| 60–69 | 2.61 | 2.63 | 2.63 |
|  | (2.03 3.35) | (2.08 3.44) | (2.10 3.20) |
| 70–80 | 4.84 | 4.84 | 4.87 |
|  | (3.80 6.18) | (3.87 6.24) | (3.87 5.91) |
| **Risk score [1]** | | | |
| Healthier patients | Ref | Ref | Ref |
| More complex patients | 2.07 | 2.11 | 2.12 |
|  | (1.90 2.27) | (1.93 2.30) | (1.93 2.33) |
| **Congestive heart failure (CHF)** | | | |
|  | 2.80 | 2.88 | 2.84 |
|  | (2.56 3.07) | (2.64 3.14) | (2.45 3.24) |
| **Country** | | | |
| Sweden | Ref | Ref | Ref |
| Denmark | 1.36 | 1.46 | 1.41 |
|  | (1.14 1.61) | (1.03 2.01) | (1.00 1.96) |
| Portugal | 1.78 | 1.83 | 1.88 |
|  | (1.55 2.05) | (1.37 2.31) | (1.43 2.42) |
| Slovenia | 1.06 | 1.01 | 1.07 |
|  | (0.83 1.36) | (0.61 1.85) | (0.66 1.80) |
| Spain | 1.36 | 1.42 | 1.38 |
|  | (1.21 1.52) | (1.14 1.73) | (1.10 1.66) |
| **Hospitals** | | | |
| MOR on patients w/o CHF |  | 1.40 | 1.42 |
| MOR on CHF patients |  |  | 1.69 |
| **Variance** | | | |
| Hospital intercept (95% credible interval) |  | 0.12 (0.08 0.18) | 0.13 (0.08 0.21) |
| CHF-mortality slope (95% credible interval) |  |  | 0.31 (0.17 0.49) |
| **Variance Partition Coefficient** | | | |
| Patients without CHF |  | 3.59 | 3.9 |
| Patients with CHF |  |  | 8.34 |
| **Goodness of fit** | | | |
| Bayesian DIC |  | 17384.88 | 17317.35 |

Models: Estimations in the basal model are obtained from single-level logistic modelling; estimations of the general contextual effect are obtained from MLRM modelling hospitals as random effect, by first just modelling a random intercept, then adding a random slope for patients with CHF. Figures represent Odds Ratios and 95% confidence intervals, except in the case of hospital estimates where MOR and confidence intervals are used. Note that, while in the random intercept model there is one value for MOR, in the model with random slope we obtain a different MOR for patients with and for patients without CHF.

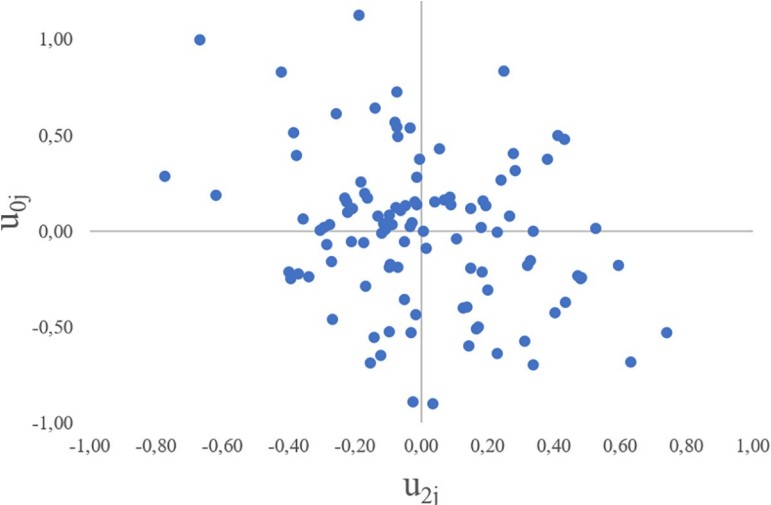

**Fig 2. Comparison of the hospital effect for the patient without CHF and the hospital effect conditioned to CHF patients.** Weighted Kappa Index -0,02 (CI95% -0,08 to 0.04).

### 3.1. Is the hospital effect consistent between estimations?

Once was the existence of hospital variance and the observation of a better goodness-of-fit of the model with random slope examined and proved, its residuals were compared. Fig 2 represents the comparison of the hospital effect for patients without CHF [$u_{oj}$] and the hospital effect conditioned to the coexistence of CHF [$u_{2j}$].

Once hospitals were classified in accordance with their level of performance (high, moderate or low), the agreement in the classification of hospital performance was non-existent [weighted Kappa Index value of -0,02 (CI95% -0,08 to 0.04)] suggesting a distinct performance in both groups of patients.

## 4. Discussion

Assuming the construct validity of AMI case-fatalities as a measure of hospital quality, this performance assessment exercise, based on 46,875 hospital admissions from five countries, shows that hospital outcomes differ when it comes to specific subgroups of patients, in our case, patients with CHF. Indeed, the greater MOR for the model including a random slope (i.e. assuming an interaction term for patients with CHF) reveals a greater influence of "hospital of treatment" when it comes to the case mortality rates for CHF patients (from MOR 1.42 in patients without CHF to MOR 1.69 with CHF).

Finally, the lack of correlation between the hospital effects on the non-CHF AMI patients and AMI patients with CHF (weighted Kappa Index = -0.02), prompts the need for analysing hospital effects on regular and specific subgroups of patients.

### 4.1. Caveats with regard to the lack of concordance

Despite the mathematical robustness of the results in terms of goodness-of-fit of the model and precision, two questions might be challenging the lack of concordance between the hospital effect on non-CHF vs. CHF AMI patients.

We could hypothesize, for example, that systemic factors could affect the GCE estimations distinctly, if the number of CHF patients per hospital is uneven across the sample (e.g. because of biased coding practices, because more complex patients arrive at certain hospitals, or

because of differential expertise in the treatment of more fragile patients between centres). Although we have reduced this potential risk by excluding the smaller hospitals from the sample, if those phenomena are true they could have an influence on the estimations of the hospital contextual effect in the specific subgroup of patients, resulting in a higher risk of death associated to the hospital of treatment in those hospitals with more CHF patients. Fig 3 showing the potential correlation between the prevalence of high-risk patients (x axis) and the estimated risk in terms of $u_{2j}$(y axis) shows that this is not the case for the hospitals in the sample, ruling out this possibility.

Another point that could eventually affect the hospital contextual effect differently on non-CHF vs. CHF patients is the surviving bias in those with no concurrent CHF. Indeed, AMI patients with concomitant CHF (most of them STEMI cases) are supposed to be more likely to die within the first 24 hours. In these cases, patients might die in the emergency room. After analysing the survival curves for both groups of patients, the negligible differences observed in the first 24 hours after in-hospital admission strongly suggest that under-recording is not likely happened in our sample [S2 Appendix shows survival curves for each country]. However, as patients who died in the emergency room are not part of our dataset, we cannot discard some under-representation in those CHF patients. Whether this fact could imply any bias in the estimation is unknown.

## 4.2. Implication of the use of random-slope MLRM in hospital performance assessment

In contrast with single level estimations, MLRM takes into account the multilevel structure of the variance existing in the data (e.g. patients nested within hospitals), accounting for the interdependence of patient outcomes within a hospital and allowing a less biased estimation of uncertainty, providing weighted estimations of average hospital risk (i.e. shrunken residuals) and allowing a more reliable assessment of the units under study.

As compared to GEE, both MLRM and GEE assume the existence of a GCE assessing hospital performance. This contextual effect is termed "general" because it reflects the influence of the hospital context as a whole, without specifying any contextual characteristics other than the very boundaries that delimit the hospital [36]. This GCE expresses the joint effect of an array of factors like, for instance, the skills and specialization of the physicians, the available access to adequate technology as well as the quality of treatment and care in the hospital. In

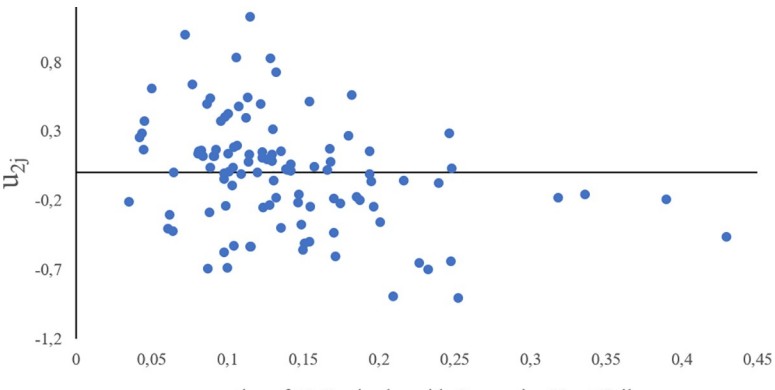

**Fig 3. Correlation between the prevalence of CHF patients and the hospital effect conditioned to those patients (u2j).** This graph contrasts the possibility of a higher hospital contextual effect due to the higher prevalence of CHF patients admitted in the hospitals in the sample. No positive correlation is observed.

such a way, the hospital context may condition patient outcomes beyond individual characteristics; that is, the same patient might have a different outcome if he or she is treated in a different hospital. However, while GEE modelling takes for granted that this GCE can be quantified by measuring differences between hospital averages only, in MLRM the GCE is measured by the share of the total patient variance that is between hospital averages; that is, the MLRM does not dislocate the individual patients from the hospital averages, but rather considers that there is a distribution of individual outcomes that can be decomposed into two levels of analysis, the individual and the hospital [9, 12, 14]. Therefore, "hospital effects" (i.e. GCE) are not properly appraised by studying the differences between hospital averages alone, but by quantifying the share of the total patient heterogeneity (i.e. variance) that exists at the hospital level [37–39]. To do this, MLRM estimates the hospital variance and its derivatives partition coefficient as a measure of the hospital GCE. Thus, when studying a specific quality outcome in patients from different hospitals, the higher the hospital variance, the more relevant is the hospital context to understanding the differences between patient outcomes [12].

More importantly, unlike other methods used to analyse clustered information (i.e. patients nested within hospitals) MLRM considers individual-level associations to be hospital-specific and drops the assumption that individual level associations are the same for all the hospitals. Consequently, in an extended MLRM with RS, hospital variance, and thereby hospital GCE, becomes a function of the patients' heterogeneity. In other words, by including a RS for a specific subgroup of patients, the hospital effect is not just a function of the very boundaries of the hospital but also a function of patients' features of interest (i.e. in our case having CHF). In practice, for a dichotomous variable, we obtain a hospital variance (i.e. a GCE) for patients without CHF and a different hospital variance for patients with CHF. This becomes, beyond considerations of interpretation, the analytical advantage of MLRM as opposed to GEE modelling.

### 4.3. Implications for hospital performance reporting

Some authors have already suggested, while acknowledging the risk of using indirect standardization in hospital performance assessment [20, 22] or in the context of social epidemiology [23], that not considering patient heterogeneity could lead to an inappropriate assessment of performance. This paper empirically underpins the need for exploring both the hospital effect for patients with or without CHF.

Therefore, a clear message is conveyed to those interested in the public reporting of performance measures. Beyond the assumption that performance assessment using administrative data is not a firm diagnostic tool but rather an instrument for screening, reporting mechanisms, more specifically league tables or funnel plots, [9, 18] should represent hospital performance according to the results of the MLRM. If the model without a random slope prevails (which is not the case in our example), a single representation for the average patient might be enough; however, if a MLRM with random slope better explains the difference in hospital outcomes, then public reporting should represent hospital effects separately for specific subgroups of patients.

One last important implication for decision-makers is that MLRM provides a measure of the effect of size (i.e. to what extent the hospital contextual effect is relevant to the differences in health outcomes) through a number of statistics (hospital variance, variance partition coefficients, and MOR) not yielded by the popular indirect standardization methods or the GEE models. This feature makes MLRM findings more actionable than other approaches.

## 5. Conclusions

The hospital contextual effect in 107 hospitals from five different European countries was different in non-CHF AMI patients and AMI patients with CHF, suggesting that accounting for

patient heterogeneity should be a requirement for adequately characterising and reporting hospital performance.

MLRM is flexible enough to allow the joint analysis of both overall effects and patient-specific hospital effects, providing accurate estimations of performance as well as a measure of the actual relevance of the hospital contextual effect.

## Supporting information

**S1 Appendix. Description of selected codes.** Inclusion and exclusion criteria and codes for episode selection.
(DOCX)

**S2 Appendix. Survival curves testing differential underreporting in CHF patients.**
(DOCX)

## Acknowledgments

We are indebted to Mircha Poldrugovac as a member of the Slovenian team in ECHO and Natalia Martínez-Lizaga for her support in the extraction and preparation of data from the ECHO dataset for the purposes of this study, and also to the Spanish Health Authorities who granted access to the hospital datasets.

ECHO Consortium: IACS's team (Bernal-Delgado E, García-Armesto S, Martínez-Lizaga M, Comendeiro-Maaløe M, Seral-Rodríguez M, Estupiñán-Romero F, Angulo-Pueyo E, Ridao-López M, and Baixaulí C, Librero J as affiliated researchers), University of Southern Denmark's team (Christiansen T, Thygesen LC), University of Nova Lisboa's team (Mateus C, Nunes C, Joaquim I), National Institute of Public Health of Ljubljana's team (Yazbeck AM, Galsworthy M, Albreht T), UMIT's team (Munck J, Güntert B) and EHMA's team (Bremmer J, Giepmans P, Dix O).

## Author Contributions

**Conceptualization:** Micaela Comendeiro-Maaløe, Juan Merlo, Enrique Bernal-Delgado.

**Data curation:** Micaela Comendeiro-Maaløe, Francisco Estupiñán-Romero, Juan Merlo.

**Formal analysis:** Micaela Comendeiro-Maaløe.

**Investigation:** Francisco Estupiñán-Romero, Lau Caspar Thygesen, Céu Mateus, Enrique Bernal-Delgado.

**Methodology:** Micaela Comendeiro-Maaløe, Juan Merlo, Enrique Bernal-Delgado.

**Project administration:** Enrique Bernal-Delgado.

**Resources:** Francisco Estupiñán-Romero, Lau Caspar Thygesen, Céu Mateus.

**Supervision:** Juan Merlo, Enrique Bernal-Delgado.

**Validation:** Micaela Comendeiro-Maaløe, Lau Caspar Thygesen, Céu Mateus, Juan Merlo, Enrique Bernal-Delgado.

**Visualization:** Micaela Comendeiro-Maaløe, Francisco Estupiñán-Romero.

**Writing – original draft:** Micaela Comendeiro-Maaløe, Enrique Bernal-Delgado.

**Writing – review & editing:** Micaela Comendeiro-Maaløe, Francisco Estupiñán-Romero, Lau Caspar Thygesen, Céu Mateus, Juan Merlo, Enrique Bernal-Delgado.

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
