## [Decision Letter · Decision Letter 0]

19 Nov 2019

PONE-D-19-28081

Acknowledging the role of patient heterogeneity in hospital outcomes reporting: mortality after acute myocardial infarction in five European countries.

PLOS ONE

Dear Dr. Comendeiro-Maaløe ,

Thank you for submitting your manuscript to PLOS ONE. After careful consideration, we feel that it has merit but does not fully meet PLOS ONE’s publication criteria as it currently stands. Therefore, we invite you to submit a revised version of the manuscript that addresses the points raised during the review process.

We would appreciate receiving your revised manuscript by 60 days. To enhance the reproducibility of your results, we recommend that if applicable you deposit your laboratory protocols in protocols.io, where a protocol can be assigned its own identifier (DOI) such that it can be cited independently in the future. For instructions see: http://journals.plos.org/plosone/s/submission-guidelines#loc-laboratory-protocols

We look forward to receiving your revised manuscript.

Kind regards,

Timir Paul

Academic Editor

PLOS ONE

Journal Requirements:

4. Your ethics statement must appear in the Methods section of your manuscript. If your ethics statement is written in any section besides the Methods, please move it to the Methods section and delete it from any other section. Please also ensure that your ethics statement is included in your manuscript, as the ethics section of your online submission will not be published alongside your manuscript.

5.  Thank you for stating in your Funding Statement:

"The ECHO Project received funding from the 7th Framework Programme of the European Union (2010-2014) with Grant HEALTH-F3-2010-242189 and from the European Union’s Health Programme (2014-2020) with Grant 664691/BRIDGE Health.

On the other hand, the authors Enrique Bernal-Delgado, Francisco Estupiñán-Romero and Micaela Comendeiro-Maaløe have been partially funded by the Institute for Health Carlos III through a public competitive grant (RD16/0001/0007) as part of the network for Health Services Research on Chronic Patients (REDISSEC)

i) Please provide an amended statement that declares *all* the funding or sources of support (whether external or internal to your organization) received during this study, as detailed online in our guide for authors at http://journals.plos.org/plosone/s/submit-now.  Please also include the statement “There was no additional external funding received for this study.” in your updated Funding Statement.

ii) Please include your amended Funding Statement within your cover letter. We will change the online submission form on your behalf.

"The ECHO Project received funding from the 7th Framework Programme of the European Union (2010-2014)

with grant HEALTH-F3-2010-242189 and from the European Union’s Health Programme (2014-2020) with grant

664691/BRIDGE Health."

"The ECHO Project received funding from the 7th Framework Programme of the European Union (2010-2014) with Grant HEALTH-F3-2010-242189 and from the European Union’s Health Programme (2014-2020) with Grant 664691/BRIDGE Health.

On the other hand, the authors Enrique Bernal-Delgado, Francisco Estupiñán-Romero and Micaela Comendeiro-Maaløe have been partially funded by the Institute for Health Carlos III through a public competitive grant (RD16/0001/0007) as part of the network for Health Services Research on Chronic Patients (REDISSEC)

The funders had no role in study design, data collection and analysis, decision to publish, or preparation of the manuscript".

8. ** Please include your tables as part of your main manuscript and remove the individual files **. Please note that supplementary tables (should remain/ be uploaded) as separate "supporting information" files.

Reviewers' comments:

Reviewer's Responses to Questions

**Comments to the Author**

1. Is the manuscript technically sound, and do the data support the conclusions?

Reviewer #1: Yes

Reviewer #2: Yes

2. Has the statistical analysis been performed appropriately and rigorously? 

Reviewer #1: Yes

Reviewer #2: Yes

3. Have the authors made all data underlying the findings in their manuscript fully available?

Reviewer #1: Yes

Reviewer #2: Yes

4. Is the manuscript presented in an intelligible fashion and written in standard English?

Reviewer #1: Yes

Reviewer #2: Yes

5. Review Comments to the Author

Reviewer #1: The paper entitled “Acknowledging the role of patient heterogeneity in hospital outcomes reporting:

mortality after acute myocardial infarction in five European countries.” by Comendeiro-Maaløe, et al. assesses the relevance of accounting for patients’ heterogeneity when reporting on hospitals’ performance.

The present study suggests that accounting for patients’ heterogeneity is required to adequately characterize and report on hospitals performance.

The topic of this manuscript is relevant to the field of this journal, and

makes a significant contribution to this field. Tables are beautiful and

easy to understand.

This article is concise, well-written, and educational for readers of this

journal. I would advise some minor changes:

The introduction is difficult to follow and needs to be simplified for the average reader. For instance, Studies assessing the effect of Post MI heterogeneities have largely focused on mortality as the primary outcome. Differences in post MI survival can be explained by disparities in available resources and medical care.

The manuscript is fraught with some minor grammatical errors and needs to be thoroughly proofread before submission.

Flowchart of patient participation, exclusion and inclusion should be included in the manuscript if possible

Reviewer #2: This is an observation study which evaluates the hospital effect in 107 hospitals from five different European countries in non-CHF AMI patients and AMI patients with CHF. Recognizing the importance of patient heterogeneity is important in reporting patient outcomes and assessing hospital performance. Different hospital vary in their underlying risk of any event. In this article by Comendeiro-Maaløe et al., multilevel regression model has been used for hospital-specific interpretations. The study assess two consecutive endpoints including variation in the hospital effect and level of agreement in hospital outcomes. They find that accounting for patients’ heterogeneity should be required to adequately characterize and report hospitals performance. Overall this paper is well written, tables and figures have been presented nicely. I would suggest editor accept this manuscript.

6. PLOS authors have the option to publish the peer review history of their article (what does this mean?). If published, this will include your full peer review and any attached files

---

## [Author Response · Author response to Decision Letter 0]

14 Jan 2020

(Hereby we expose the same information included in the indexed document "response to reviewers")

Ref: PONE-D-19-28081-R1

Responses to the first round of reviews

Version 2020 01 10

Dear editor,

Thank you very much for allowing us to submit a reviewed version of our manuscript. Please, find below our replies to your concerns and the first reviewer’s comments in a case-by-case manner. Those points that implied a change in the manuscript have been conveniently highlighted.

Editor’s concerns and journal’s requirements:

2. We note that you have indicated that data from this study are available upon request. PLOS only allows data to be available upon request if there are legal or ethical restrictions on sharing data publicly. In your revised cover letter, please address the following prompts:

a) If there are ethical or legal restrictions on sharing a de-identified data set, please explain them in detail (e.g., data contain potentially identifying or sensitive patient information) and who has imposed them (e.g., an ethics committee). Please also provide contact information for a data access committee, ethics committee, or other institutional body to which data requests may be sent (We will update your Data Availability statement on your behalf to reflect the information you provide).

Response: As required, we have added an informative paragraph in the revised cover letter as follows:

“The access to pseudoanonimised patient-level administrative data (patient episodes: hospital admissions and discharges) supporting the manuscript findings is restricted to avoid data misuse in accordance with the System Level Security Policy of the Unit for Health Services and Policy Research at the Institute for Health Sciences in Aragon (IACS) (Available at: www.atlasvpm.org/grupo-coordinador/ARiHSPinformationsecuritypolicy) within the framework of the Spanish legal system, in particular, Regulation (EU) 2016/679, the Law 3/2018 on Personal Data Protection, the Law 14/2007 on Biomedical Research and the Law 37/2007 and Law 18/2015 on the Public Sector Information Reuse. Consequently, we have specified that only aggregated data may be accessed, upon request, throughout our data sharing agreement, by contacting the senior author of the manuscript Enrique Bernal-Delgado (ebernal.iacs@aragon.es) and/or the legal officer Ramón Launa-Garcés (rlaunag.iacs@aragon.es)”

5. Thank you for stating in your Funding Statement:

"The ECHO Project received funding from the 7th Framework Programme of the European Union (2010-2014) with Grant HEALTH-F3-2010-242189 and from the European Union’s Health Programme (2014-2020) with Grant 664691/BRIDGE Health. On the other hand, the authors Enrique Bernal-Delgado, Francisco Estupiñán-Romero and Micaela Comendeiro-Maaløe have been partially funded by the Institute for Health Carlos III through a public competitive grant (RD16/0001/0007) as part of the network for Health Services Research on Chronic Patients (REDISSEC). The funders had no role in study design, data collection and analysis, decision to publish, or preparation of the manuscript."

i) Please provide an amended statement that declares *all* the funding or sources of support (whether external or internal to your organization) received during this study, as detailed online in our guide for authors at http://journals.plos.org/plosone/s/submit-now. Please also include the statement “There was no additional external funding received for this study.” in your updated Funding Statement.

ii) Please include your amended Funding Statement within your cover letter. We will change the online submission form on your behalf.

Response: As required, we have added an amended statement in the revised cover letter as follows:

“The ECHO Project received funding from the 7th Framework Programme of the European Union (2010-2014) with Grant HEALTH-F3-2010-242189 and from the European Union’s Health Programme (2014-2020) with Grant 664691/BRIDGE Health. Additionally, the authors Enrique Bernal-Delgado, Francisco Estupiñán-Romero and Micaela Comendeiro-Maaløe have been partially funded by the Institute for Health Carlos III through a public competitive grant (RD16/0001/0007) as part of the network for Health Services Research on Chronic Patients (REDISSEC). The funding sources received during this study played no role in the study design, data collection or the analysis, decision to publish, or preparation of the manuscript. There was no additional external funding received for this study”

"The ECHO Project received funding from the 7th Framework Programme of the European Union (2010-2014) with grant HEALTH-F3-2010-242189 and from the European Union’s Health Programme (2014-2020) with grant 664691/BRIDGE Health."

"The ECHO Project received funding from the 7th Framework Programme of the European Union (2010-2014) with Grant HEALTH-F3-2010-242189 and from the European Union’s Health Programme (2014-2020) with Grant 664691/BRIDGE Health. On the other hand, the authors Enrique Bernal-Delgado, Francisco Estupiñán-Romero and Micaela Comendeiro-Maaløe have been partially funded by the Institute for Health Carlos III through a public competitive grant (RD16/0001/0007) as part of the network for Health Services Research on Chronic Patients (REDISSEC). The funders had no role in study design, data collection and analysis, decision to publish, or preparation of the manuscript".

Response: all the funding information has been removed from the manuscript as required. The funding information has also been updated and included in the revised cover letter as required in point 5.

Reviewer’s concerns:

Reviewer #1

The introduction is difficult to follow and needs to be simplified for the average reader. For instance, Studies assessing the effect of Post MI heterogeneities have largely focused on mortality as the primary outcome. Differences in post MI survival can be explained by disparities in available resources and medical care.

Response: Thank you for your suggestion, although we are afraid that by using other terms as you propose in your example, the meaning of the conceptualisation of what we methodologically develop later in the paper could be changed. Nevertheless, we believe that the corrections made in the introduction once the manuscript has been proofread by a native speaker, may ease comprehension.

The manuscript is fraught with some minor grammatical errors and needs to be thoroughly proofread before submission.

Response: thank you for the suggestion. As mentioned before, the manuscript has been proofread and corrected by a native speaker.

Flowchart of patient participation, exclusion and inclusion should be included in the manuscript if possible

Response: Thank you for the suggestion. A flowchart has been included at the beginning of the methods section, now named as figure 1.

---

## [Editor Report · Decision Letter 1]

15 Jan 2020

Acknowledging the role of patient heterogeneity in hospital outcome reporting: mortality after acute myocardial infarction in five European countries.

PONE-D-19-28081R1

Dear Dr. Enrique Bernal-Delgado,

We are pleased to inform you that your manuscript has been judged scientifically suitable for publication and will be formally accepted for publication once it complies with all outstanding technical requirements.

With kind regards,

Timir Paul

Academic Editor

PLOS ONE

Additional Editor Comments (optional):

All the reviewers' comments have been answered.
---

## [Editor Report · Acceptance letter]

21 Jan 2020

PONE-D-19-28081R1 

Acknowledging the role of patient heterogeneity in hospital outcome reporting: mortality after acute myocardial infarction in five European countries. 

Dear Dr. Bernal-Delgado:

I am pleased to inform you that your manuscript has been deemed suitable for publication in PLOS ONE. Congratulations! Your manuscript is now with our production department. 

With kind regards,

on behalf of

Dr. Timir Paul 

Academic Editor

PLOS ONE